# Discovery of E3 Ligase Ligands for Target Protein Degradation

**DOI:** 10.3390/molecules27196515

**Published:** 2022-10-02

**Authors:** Jaeseok Lee, Youngjun Lee, Young Mee Jung, Ju Hyun Park, Hyuk Sang Yoo, Jongmin Park

**Affiliations:** 1Department of Chemistry, Kangwon National University, Chuncheon 24341, Korea; 2Department of Chemistry and Biochemistry, University of California, San Diego, CA 92093, USA; 3Kangwon Radiation Convergence Research Support Center, Kangwon National University, Chuncheon 24341, Korea; 4Kangwon Institute of Inclusive Technology, Kangwon National University, Chuncheon 24341, Korea; 5Department of Biomedical Science, Kangwon National University, Chuncheon 24341, Korea

**Keywords:** target protein degradation, PROTAC, E3 ligase ligand

## Abstract

Target protein degradation has emerged as a promising strategy for the discovery of novel therapeutics during the last decade. Proteolysis-targeting chimera (PROTAC) harnesses a cellular ubiquitin-dependent proteolysis system for the efficient degradation of a protein of interest. PROTAC consists of a target protein ligand and an E3 ligase ligand so that it enables the target protein degradation owing to the induced proximity with ubiquitin ligases. Although a great number of PROTACs has been developed so far using previously reported ligands of proteins for their degradation, E3 ligase ligands have been mostly limited to either CRBN or VHL ligands. Those PROTACs showed their limitation due to the cell type specific expression of E3 ligases and recently reported resistance toward PROTACs with CRBN ligands or VHL ligands. To overcome these hurdles, the discovery of various E3 ligase ligands has been spotlighted to improve the current PROTAC technology. This review focuses on currently reported E3 ligase ligands and their application in the development of PROTACs.

## 1. Introduction

Proteins are the basic machinery of the cellular system and execute genetically programmed behaviors for cellular survival. Around 20,000 proteins have been identified from human cells and their balance in the protein network is extremely important to maintain the healthy status of cells. The dysfunction or breakdown of a single protein could lead to the disease status of cells. Therefore, discovering the small molecules that are modulating the dysregulated protein is the key strategy for the drug discovery process. However, only a limited number of proteins has been reported as druggable proteins and traditional drug discovery has been focused on those druggable proteins. Considering the fact that a numerous number of proteins in cells are essential for cellular homeostasis, targeting undruggable proteins could be a solution for the treatment of incurable diseases.

Targeted Protein Degradation (TPD) is an emerging therapeutic strategy which is considered a solution to overcome the limitations of conventional drug discovery. TPD is a powerful chemical biology approach for inducing the degradation of proteins known as undruggable, and it has had a tremendous impact on the field of recent drug discovery. TPD manipulates the E3 ligase to selectively degrade the protein of interest (POI) via the Ubiquitin Proteasome System (UPS), an intracellular proteolysis mechanism. A targeted protein degrader consists of a ligand that binds to E3 ligases, a ligand that targets the POI, and a chemical linker between the ligands. These degraders are also called proteolysis-targeting chimeras (PROTACs), and the induced proximity between the target protein and the E3 ligase promotes the ubiquitination and proteasomal degradation of the target protein. Along with PROTAC, a protein degradation strategy using the protein tag has been developed to understand the inherent function of proteins. The additional aminoacidic signal sequences such as dTAG, AiD, and SMASh Tag induce proximity-based POI ubiquitination for its degradation via the ubiquitin–proteasome system. The development of tag-based strategies is well documented in the following references [1,2,3].

Even though there are over 600 types of E3 ligases in human cells, only a very limited number of E3 ligases have been used for PROTAC technology (CRBN, VHL, IAP, and MDM2). The majority of PROTACs developed so far have been restricted to CRBN or VHL, which are ubiquitously expressed in the human body. However, the recent emergence of drug resistance for CRBN- or VHL-based PROTACs strongly suggested that the discovery of various E3 ligase ligands is highly demanding to fully exploit the PROTAC strategy [4,5,6]. In addition, considering that a number of E3 ligases have been shown to overexpress in specific types of cells or tissues—for example, brain (FBXL16, KCTD8), pancreas (ASB9), skeletal muscle (KLHL40, KLHL41), testis (DCAF4L1), fallopian tube (DCAF8L1)—harnessing new E3 ligases may offer better opportunities for PROTACs having a higher selectivity and specificity for the efficient disease treatment. In this context, an overview of the PROTAC design strategies is highly necessary to develop efficient protein degraders. This review focuses on the currently reported E3 ligase ligands and their application in the development of PROTACs.

## 2. CRBN Ligands

In the 1950s, thalidomide (**1**) was first developed by Grünenthal as a sedative for morning sickness in pregnant women. However, thalidomide was withdrawn from the market due to its severely teratogenic effects in the early 1960s [7]. During the following decades, thalidomide was further studied extensively, redeveloped as a promising immunomodulatory imide drug (IMiD), and approved for the treatment of erythema nodosum leprosum (ENL) and multiple myeloma. The thalidomide derivatives, such as pomalidomide and lenalidomide, have shown an excellent efficacy in the treatment of multiple myeloma with their immunomodulatory activities (Figure 1) [8,9,10]. However, the mechanism of action of thalidomide had not been elucidated until Ito et al. revealed that the target protein of thalidomide is cereblon, a subunit of the E3 ubiquitin ligase CUL4-RBX1-DDB1-CRBN(CRL4^CRBN^), in 2010 [11]. They demonstrated a thalidomide–CRBN complex-induced teratogenic effect in in vivo models. In addition to thalidomide, the target protein of pomalidomide and lenalidomide was identified as CRBN by Zhe et al. in 2011 [12]. The antimultiple myeloma activity of pomalidomide (**2**) and lenalidomide (**3**) was CRBN expression dependent.

In 2014, Fischer et al. presented the cocrystal structure of the DDB1–CRBN complex bound to thalidomide, lenalidomide, and pomalidomide. This research showed that CRBN is the substrate receptor of CRL4^CRBN^ and binds to thalidomide derivatives in an enantioselective manner. They further revealed that the thalidomide–CRBN complex recruits IKZF1 or IKZF3 and induces their degradations [13]. Thalidomide derivatives are composed of the phthalimide and glutarimide group (Figure 2A). The cocrystal structure (PDB code: 4CI1) of the thalidomide and CRBN complex showed that the glutarimide group of thalidomide derivatives plays an important role in CRBN binding via two major interactions: (1) the H-bond between carbonyl and amide groups of the glutarimide group and His380 and Trp382 of CRBN; and (2) van der Waals interactions between the glutarimide group and the hydrophobic pocket of CRBN composed of Phe404, Trp388, and Trp402 (Figure 2B). A carbonyl group of the phthalimide group also contributes a H_2_O-mediated hydrogen bonding with the His359 of CRBN. This cocrystal structure revealed that the solvent-exposed site of thalidomide is the benzene ring of the phthalimide group, which can be further conjugated for the PROTAC design without a loss of the binding affinity [14]. Based on this structural information, the development of thalidomide derivatives has been able to be accelerated.

In 2015, the Crews and Bradner group reported that bromodomain-containing protein 4 (BRD4) targets PROTACs. **ARV-825** (DC_50_ < 1 nM) [15] and **dBET1** (EC_50_ = 430 nM) were synthesized by conjugating a BRD4 ligand, JQ1, with a thalidomide derivative for CRBN engagement (Figure 3). Both PROTACs effectively triggered the degradation of BRD4 in the cells and inhibited the cell proliferation. **dBET1** showed an excellent antiacute myeloid leukemia (AML) efficacy in vitro and in vivo [16]. After **dBET1** was reported as the first PROTAC in vivo, various thalidomide derivatives have been explored to improve the efficiency of the PROTAC technique.

In 2018, the Crew group reported 22 thalidomide analogs with a rapid one-pot synthesis without purification. They measured the binding affinities of newly synthesized thalidomide analogs to CRBN with a surface plasmon resonance analysis. In addition, the ability to induce the degradation of Aiolos and CK1α was investigated. Among them, three thalidomide analogs (**4**, **5**, **6**) showed improved pharmacological properties and good CRBN binding affinities (*K*_D_ for **4**, **5**, and **6** were 55, 549, and 111 nM, respectively). They found that the chemical modification of the phthalimide part in thalidomide did not induce a significant deterioration in the CRBN binding affinities [17]. In 2019, the Yang group synthesized PROTAC, **DGY-08-097**, which induces hepatitis C virus (HCV) NS3/4A protease degradation by conjugating **6** with Telaprevir, an FDA-approved drug for the treatment of HCV [18] (IC_50_ = 247 nM, DC_50_ = 50 nM). **DGY-08-097** showed antiHCV activity in the cellular infection model.

C4 Therapeutics also filed a patent for the synthesis of CRBN-targeting moieties (Degrons) that can be conjugated to target protein ligands. They reported various piperidine-2,6-dione derivatives by conjugating O- and N-linked heterocycles. A fluorescence polarization (FP) assay demonstrated that **7** was one of the most promising CRBN ligands. The degrons are capable of functioning as molecular glues and downregulating the levels of the Aiolos or Ikaros protein, which can lead to the treatment of leukemia, acute myeloid leukemia, chronic lymphoblastic leukemia, and multiple myeloma just like other IMiDs. They conjugated the developed E3 ligase ligand to JQ1 and named them **Degronimers**. The representative **Degronimers 1** and **2** showed an excellent binding affinity to CRBN [19] (*K*_D_ < 10 μM). They also reported various SMARCA2 degraders using the degrons. The N-linked degrons, **8**, was developed as PROTAC (**compound 156**) for SMARCA2 degradation with a nanomolar efficacy through a HiBiT degradation assay (DC_50_ = 3 nM) [20].

The Hwang group designed aminobenzotriazino glutarimides as novel CRBN ligands and discovered **TD-106**
**(9)**. The **TD-106**
**(9)** exhibited a better degradation efficiency of IKZF1/3 than that of pomalidomide in NCI-H929 cells. After the confirmation of **TD-106**
**(9)** as a direct CRBN binder through a thermal stability shift, an in vivo xenograft model study demonstrated that intraperitoneally injected **TD-106**
**(9)** showed antitumor activity after 14 days of administration [21]. They further synthesized a BRD4-targeting PROTAC, **TD-428** (DC_50_ = 0.32 nM), by conjugating JQ1 and **TD-106**
**(9).** The successful BRD4 degradation in 22Rv1 cells induced by **TD-428** was confirmed. They also reported an androgen receptor (AR) degrader, **TD-802** (DC_50_ = 12.5 nM), for the treatment of metastatic castration-resistant prostate cancer using the **TD-106**
**(9)** ligand [22].

In 2019, Arvinas unveiled that the **ARV-110** (DC_50_ ~ 1 nM) and **ARV-471** (DC_50_ ~ 1 nM) are targeting AR and estrogen receptor (ER), respectively. Both degraders have been spotlighted since they are currently in phase two of clinical trials [23,24]. **ARV-110** was synthesized via the conjugation of a thalidomide derivative (**4**) and AR ligand. **ARV-471** was developed via the conjugation of a thalidomide derivative (**5**) and ER ligand. Both shared the same linker structure for the conjugation of CRBN and target protein ligands. The most interesting feature of these PROTACs is that both can be administered orally, which was not easily achievable in other PROTACs.

Kymera Therapeutics reported various CRBN ligands having two rings conjugated with diverse linkers in 2019. They measured their affinities toward CRBN through a time-resolved fluorescence resonance energy transfer (TR-FRET) assay [25]. They synthesized a PROTAC, **I-265** (DC_50_ < 0.1 μM), by conjugating **10** and an interleukin-1 receptor activated kinase (IRAK) inhibitor. They showed that **I-265** degraded the IRAK4 protein in human peripheral blood mononuclear cells (PBMCs) [26]. One of their ligands, **SB572027 (11)**, was used for the synthesis of BTK-targeting PROTACs by the Chinese biotechnology company Beigene. Beigene synthesized a series of PROTACs that conjugated a BTK inhibitor with various thalidomide derivatives capable of recruiting the target protein to the E3 ubiquitin ligase. They performed an ELISA assay for BTK detection to evaluate the activity of a series of the PROTACs. Among them, **compound 155** using **SB57027 (11)** as a CRBN ligand showed nanomolar degradation activity (DC_50_ = 7.2 nM) [27].

In 2021, the Rankovic group reported highly stable CRBN binders and their application for PROTACs. They replaced the hydrolysis-labile phthalimide moieties of thalidomide with phenyl groups to synthesize phenyl glutarimide (**PG, 12**). **PG (12**, IC_50_ = 2.191 μM) showed outstanding stability (t_1/2_ > 24 h) compared with thalidomide (t_1/2_ = 3.3 h) in cell media without affecting the binding affinity with CRBN (IC_50_ for thalidomide = 1.282 μM, lenamide = 0.699 μM, and pomalidomide = 0.4 μM). **PG (12)** also showed superior metabolic stability in mouse and human liver microsomes. A **PG**-based PROTAC (**SJ995973**, DC_50_ = 0.87 nM) showed potent antiproliferative efficacy in both MV4-11 (IC_50_ = 3 pM) and HD-MB03 cells (IC_50_ = 1.83 nM) [28]. In a follow-up study, the Rankovic group used **PG** (**12**) to synthesize Janus kinase (JAK)-targeting PROTACs. Through the structure activity relationship (SAR) study, they found a **PG** (**12**)-based PROTAC, **SJ10542**, with highly selective JAK degradation and reduced GSPT1 degradation compared with thalidomide-based PROTACs (JAK2 DC_50_ = 14 nM, JAK3 DC_50_ = 11 nM) [29].

In 2021, Wang et al. filed a patent about the synthesis of various CRBN ligands for PROTAC design [30]. Among their compounds, **13** and **14** were conjugated to spirocyclic AR ligands to synthesize AR-degrading PROTACs for the treatment of prostate cancer. They measured AR degradation in VCaP cells and **compound 311** showed the efficient degradation of AR with a nanomolar concentration (DC_50_ < 10 nM). Moreover, they demonstrated the superior oral bioavailability of **compound 311** [31].

In 2021, Novartis developed a BRD9 degrader for cancer treatment. BRD9 is a subunit of the SWI/SNF complex, which has been reported as a drug target for the treatment of synovial sarcoma and acute myeloid leukemia. A series of BRD9 PROTACs were synthesized via the conjugation of BRD9-targeting ligands and CRBN-targeting E3 ligase ligands with various types of short and rigid piperidine linkers. Among the synthesized PROTACs, **compound B6** and **compound E32**, based on the CRBN binders **15** and **16**, respectively, showed BRD9 DC_50_ at the nanomolar concentration [32] (DC_50_ = 1 nM for both **compound B6** and **E32**).

## 3. VHL Ligands

The von Hippel–Lindau (VHL) protein is the substrate receptor protein of the Cullin 2 E3 ligase. Hypoxia-inducible factor 1α (HIF-1α) is one of the substrate proteins of VHL. Considering the HIF-1α-mediated upregulation of proangiogenic factors, an inhibitor of VHL would increase erythropoietin and can be used for the treatment of chronic anemia and cancer chemotherapy. In 2012, the Ciulli and Crews group reported a series of small molecule inhibitors targeting VHL for the first time. They rationally designed inhibitors of the VHL/HIF-1α interaction in silico. After the discovery of their initial hit, they synthesized a focused library of hydroxy proline derivatives with solid phase synthesis. They found VHL ligand **17** with single-digit micromolar activity through an FP assay (Figure 4). The cocrystal structure of the ligand bound to VHL confirmed that the ligand mimics the binding mode of the HIF-1α to VHL [33]. They demonstrated that the major binding affinity of VHL ligands originates from the H-bond between the hydroxyl group of pyrrolidine and His115 and Ser111, as well as the interaction between the amide NH group and a carbonyl group of His110 (Figure 5A). In addition, the phenyl group, adjacent to the amide group, contributes to the high binding affinity of the VHL ligand via a π-π interaction with Tyr98.

In a follow-up study, the Ciulli group designed and optimized the initial VHL ligands based on X-ray crystal structures and reported new VHL ligands, **VH032** (**18**, *K*_D_ = 185 nM) and **19** (*K*_D_ = 291 nM), with nanomolar binding affinities, in 2014 [34]. Using **VH032** (**18**) and **19**, the Ciulli group reported bromodomain and extra-terminal domain (BET) protein-targeting PROTACs **MZ1** (*K*_D_ = 149 nM) and **MZ3** (*K*_D_ = 311 nM) (Figure 6) [35]. They observed the **MZ1**-mediated degradation of BRD2, BRD3, and BRD4 at the single digit micromolar concentration (BRD4 DC_50_ < 100 nM).

In 2015, the Crews group proposed HaloPROTAC using their VHL ligands. Based on their previously reported crystallographic evidence, they identified the possible linker positions by confirming the solvent-exposed sites of the VHL ligands. Not only the linker position but also a proper linker length were studied for VHL-mediated protein degradation. Finally, they successfully designed **HaloPROTAC3** (DC_50_ = 19 nM) by conjugating a chloroalkane linker to the VHL ligand (**20**, IC_50_ = 0.34 μM) for the degradation of the HaloTag fusion protein [36]. The Crews group revealed that the linker position of the VHL ligand largely affects the substrate specificity of PROTAC. They synthesized PROTACs with the linkers on the left-handed amide side and right-handed phenyl side of the VHL ligand, respectively (Figure 5B). The PROTAC (**SJF-6683**) conjugated a p38 MAPK ligand, foretinib, to the right-hand phenyl side of the VHL ligand (**20**) selectively and strongly degraded p38δ, a specific p38 isoform (p38δ DC_50_ = 46.17 nM) [37].

**Figure 4 molecules-27-06515-f004:**
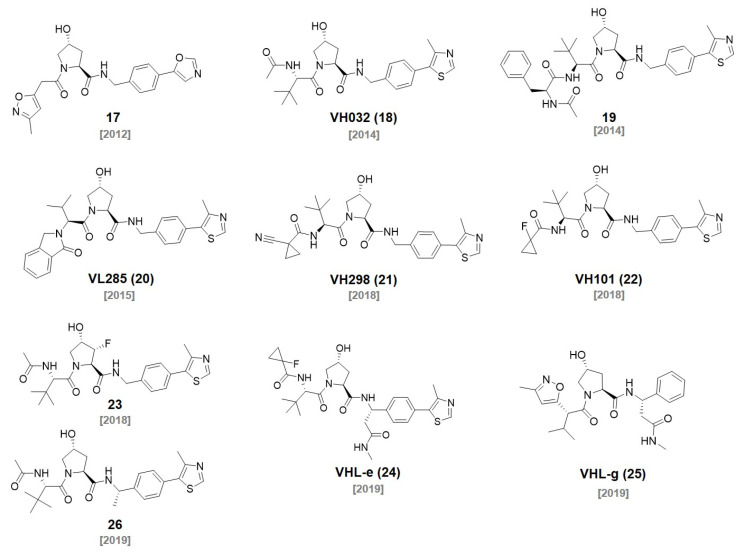
Chemical structures of reported VHL ligands during the last decade.

**Figure 5 molecules-27-06515-f005:**
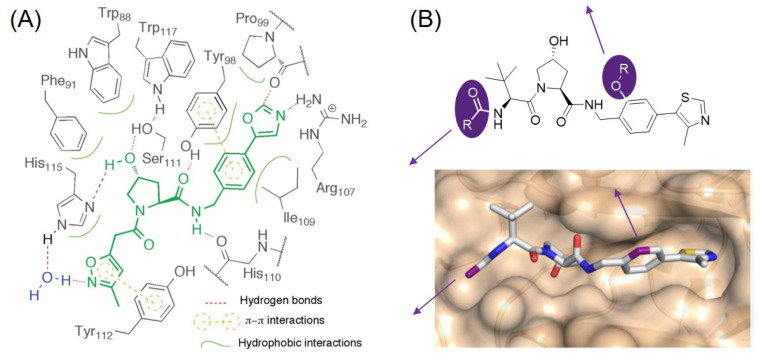
(**A**) Schematic illustrations showing the binding mode of the VHL ligand **17**. Reprinted with permission from ref. [33] (Copyright 2012 American Chemical Society). (**B**) Chemical structure of a representative VHL ligand analogue having two different linker positions. Reprinted with permission from ref. [37] (Copyright 2019 Springer Nature).

**Figure 6 molecules-27-06515-f006:**
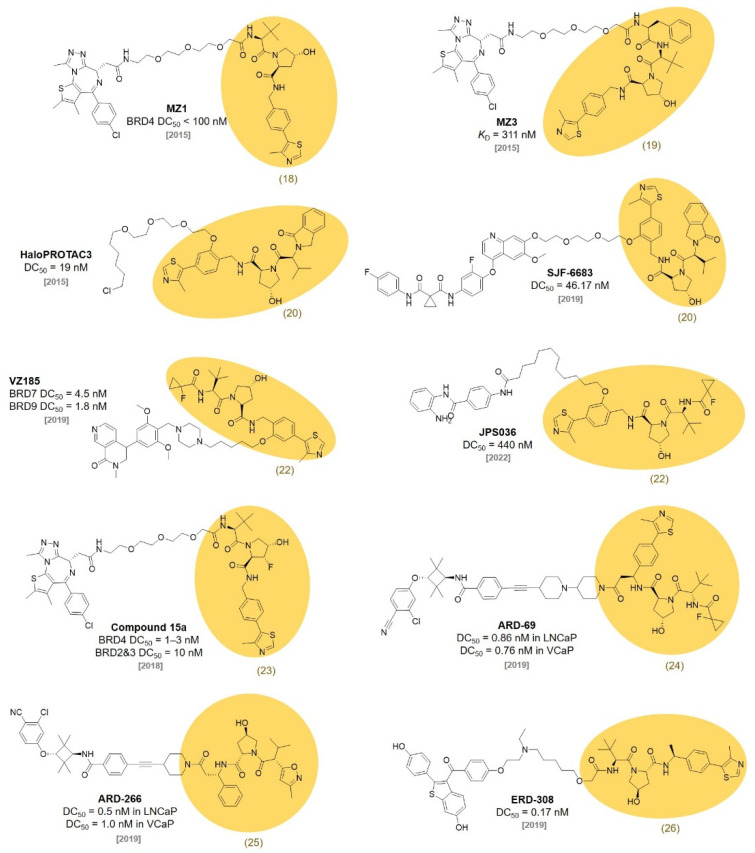
Reported PROTACs based on the VHL ligands.

In 2018, Ciulli reported the structure-guided rational optimization of **VH032 (18)**. Increasing the lipophilicity of the VHL ligands led to a higher cell permeability and higher binding affinity to the VHL protein. They discovered **VH298 (21**, *K*_D_ = 52 nM) and **VH101** (**22**, *K*_D_ = 16 nM), which showed the effective protein–protein inhibition between the VHL and HIF-1α protein. [38]. Their SAR study provided novel VHL ligands, which can be used for VHL-based PROTACs. Note that this discovery expanded the landscape of PROTAC research to other E3 ligases, which was only restricted to CRBN. **VH101** (**22**) was used in the SAR study to develop BRD7 and BRD9 degraders by the Ciulli group. A PROTAC (**VZ185**) was developed by coupling the BRD7/9 inhibitor, BI-7273, and **VH101** (**22**). **VZ185** efficiently degraded BRD7 and BRD9 simultaneously (BRD7 DC_50_ = 4.5 nM, BRD9 DC_50_ = 1.8 nM) [39].

The Hodgkinson group reported histone deacetylase (HDAC)-targeting PROTACs. They synthesized HDAC-degrading PROTACs with various linkers and VHL ligands and monitored their degradation activities. **JPS036**, a PROTAC composed of **VH101** (**22**), was developed as a selective degrader of HDAC3 with a submicromolar activity (DC_50_ = 0.44 μM) [40].

In 2018, the Ciulli group reported VHL ligands with a different stereochemistry of fluoro-hydroxyproline (F-Hyp). They synthesized four diastereoisomers of 3-fluoro-4-hydroxyproline containing VHL ligands and found that VHL can stereoselectively recognize the (3R,4S) epimer of F-Hyp (**23**). A JQ1-based PROTAC (**compound 15a**) using the (3R,4S) epimer of F-Hyp selectively degraded BRD4 at nanomolar concentrations, despite its weak affinity for VHL (*K*_D_ = 3.08 μM, BRD4 DC_50_ = 1~3 nM, BRD2 and BRD3 DC_50_ = 10 nM). This discovery was an important advance in expanding the chemical space of TPD toward low affinity molecules [41].

In 2019, the Wang group discovered new VHL ligands via the introduction of an (S)-methyl group on **VH101** based on previous work [42,43]. Through a SAR study, they found that appending an amide group to the (S)-methyl group increased the potency of the VHL ligands. The FP-based binding assay showed that **VHL-e** (**24**) binds to VHL with a high affinity (IC_50_ = 190 nM). In addition, they confirmed that the introduced stereochemistry was crucial for their binding affinity to VHL. With **VHL-e** (**24**), an effective AR degrader, **ARD-69**, was discovered after the optimization of the linker length and linking site on the ligands. **ARD-69** showed effective AR degradation activity in LNCaP (DC_50_ = 0.86 nM), VCap (DC_50_ = 0.76 nM), and 22Rv1 prostate cancer cell lines and in a VCaP xenograft mouse model [43]. The Wang group further optimized the AR-targeting PROTAC with a shorter linker length and a low-affinity VHL ligand. **ARD-266** using a weak binding VHL ligand, **VHL-g** (**25**), showed a much higher AR degradation activity than other PROTACs with higher-affinity VHL ligands (DC_50_ = 0.5 nM in LNCaP cell line, DC_50_ = 1.0 nM in VCaP cell line). Along with **compound 15a**, this study demonstrated that a low-affinity E3 ligase ligand-based PROTAC could induce the successful formation of a ternary complex with the POI for efficient degradation [44]. Subsequently, the Wang group extensively studied the SAR of an ER degrader based on a VHL ligand (**26**) and FDA-approved ER modulator, Raloxifene. As a result, a very potent ER degrader (**ERD-308**) was developed with a subnanomolar activity (DC_50_ = 0.17 nM). **ERD-308** degraded the ER and inhibited cell growth more than those of the FDA-approved selective ER degrader molecule Fulvestrant [45]. In this report, changing the VHL ligand to the CRBN ligand completely abolished the ER degradation activity of **ERD-308**.

## 4. IAP Ligands

Inhibitors of apoptosis proteins (IAPs) are the regulators of cell death and they control apoptotic events triggered by diverse stimuli. In 2007, the Vucic group developed a cellular inhibitor of the apoptosis 1 and 2 (c-IAP1 and c-IAP2) antagonist (**MV1, 27**) that binds to the baculovirus IAP repeat (BIR) domains of IAP proteins, leading to the autoubiquitination and proteasomal degradation of c-IAPs (*K*_D_ = 5.8 nM) (Figure 7). The degradation of c-IAPs by **MV1** induced TNF signaling-pathway-dependent cell death [46]. Sekine et al. reported a different cIAP1 ligand, bestatin-methyl ester (**ME-BS, 28**), which binds to the BIR3 domain of cIAP1 and induces autoubiquitination followed by the proteasomal degradation of cIAP1 [47]. Itoh et al. developed bifunctional small molecules using the two ligands described above. One of the molecules (**Compound 4b**) was designed by the conjugation of **ME-BS** (**28**) and all-trans retinoic acid (ATRA) with a polyethylene glycol (PEG) linker (Figure 8). The other was developed via the conjugation of **MV1 (27)** and ATRA (**Compound 6**) [48,49]. These compounds were found to induce the degradation of both cIAP1 and cellular retinoic acid binding protein-II (CRABPII). They named this degrader the specific and nongenetic IAP-dependent protein eraser (SNIPER). It was later utilized on other targets, such as ER [50], BRD4 [51], and BCR-ABL [52].

In 2012, Genentech discovered a potent antagonist of cIAP1/2, ML-IAP, and XIAP. The SAR study using the crystal structure led to the development of a broad spectrum IAP inhibitor, **GDC-0152 (29)**
*(K*_i_ values for XIAP-BIR3 = 28 nM, MLXBIR3SG = 14 nM, cIAP1-BIR3 = 17 nM, and cIAP2-BIR3 = 43 nM) [53]. In cocrystal structures of **GDC-0152** with ML-IAP or cIAP1, the critical interaction was the H-bond between the Asp (Asp138 for ML-IAL, Asp320 for cIAP1) and the amide group of the ligand (Figure 9). It allows for the proper positioning of the α-methyl group of the ligand to reside in the P1 cavity. In addition, another hydrophobic interaction was observed between the phenyl group of the ligand and P4 hydrophobic pocket in both crystal structures. Since most of the ligand structure is exposed to the solvent, the target protein ligands can be conjugated to diverse positions of the IAP ligands.

The Hennessy group discovered a series of aminopiperidine-based inhibitors of the IAP by mimicking the IAP binding residues of the second mitochondrial activator of caspases (Smac). They found that a bicyclic piperidine, **30** (*K*_D_ for XIAP-BIR3 = 0.9 μM), fixed in a boat form was a potent inhibitor of cIAP1 and effectively induced cIAP1 degradation in MDA-MB231 cells (EC_50_ = 5 nM) [54].

In 2013, the Cosford group reported the synthesis of a potent IAP antagonist via a highly efficient application of the Ugi four-component reaction. Their optimized IAP antagonist (**31**) showed the best binding affinity to IAPs, especially against ML-IAP (*K*_i_ = 2 nM). **31** showed a powerful anticancer activity in breast, ovarian, and prostate cell lines and had no general toxicity to noncancerous human foreskin fibroblast (HFF) cells. They performed molecular modeling to reveal key interacting residues of IAP proteins with **31** [55]. The Zheng group used **31** to synthesize IAP-recruiting BCL-X_L_ PROTACs, **compound 8A** (IC_50_ = 62 nM, DC_50_ < 500 nM). **Compound 8A** showed efficient BCL-X_L_ degradation in the T-cell lymphoma cell line, while it had reduced the human platelet toxicity [56].

In 2014, Bristol-Myers Squibb reported that **32** embedded bivalent heterodimeric IAP antagonists showed a high affinity for the BIR2 domain and an excellent IAP inhibitory activity (IC_50_ up to 3.6 nM) [57]. Pfizer synthesized **BC5P**, a PROTAC that degrades BTK using **32** (DC_50_ = 182 nM). They confirmed that **BC5P** bound to only the BIR3 domain of IAP1 but not BIR1 or BIR2 by using biolayer interferometry (BLI). They utilized molecular modeling, solution NMR, and X-ray crystallography to elucidate the structural insights of the IAP-BC5P-BTK ternary complex [58].

In 2017, the Naito group developed an ER-targeting SNIPER, **SNIPER(ER)-87**, via the conjugation of an ER ligand, 4-hydroxytamoxifen, and an IAP ligand, **LCL-161 (33)** [50]. **SNIPER(ER)-87** effectively reduced ERα protein levels at nanomolar concentrations in vitro (DC_50_ = 3 nM, IC_50_ = 15.6 nM in MCF-7 cell line, IC_50_ = 9.6 nM in T47D cell line). **SNIPER(ER)-87** showed good metabolic stability in the serum. The intraperitoneal administration of **SNIPER(ER)-87** reduced the growth of ER-positive human breast tumors in vivo. They also conjugated **LCL-161** derivatives to JQ1, a PDE4 inhibitor, and dasatinib for SNIPERs targeting BRD4, PDE4, and BCR-ABL proteins to demonstrate the usefulness of **LCL-161** derivatives for the development of various targeted protein degraders.

In 2018, the Naito group optimized their previously reported **SNIPER(ER)-87** by incorporating various **LCL-161**-derivative IAP ligands (**29**, **30**, **35**, **36**). With the improved IAP binding affinities of the **LCL-161** derivatives, the optimized SNIPER(ER)s exhibited better binding affinities toward cIAP1, cIAP2, and XIAP. However, the E3 ligase binding affinities of SNIPER(ER)s were not exactly correlated to their target protein degradation efficiencies. The **GDC-0152** (**29**)-based **SNIPER(ER)-131** did not efficiently degrade ERα, despite its higher IAPs affinity than that of **SNIPER(ER)-87** (**SNIPER(ER)-131**: ERα DC_50_ > 33.8 nM, IC_50_ = 80 nM, **SNIPER(ER)-87**: ERα DC_50_ < 3 nM, IC_50_ = 110 nM). **30**-based **SNIPER(ER)-118** had a low ERα degradation efficiency compared to the original compound **SNIPER(ER)-87** (**SNIPER(ER)-118**: ERα DC_50_ > 100 nM, IC_50_ = 230 nM). **SNIPER(ER)-110** and **SNIPER(ER)-126** were the most potent ERα degraders with the lowest DC_50_ (< 3 nM) among the SNIPER(ER)s (**SNIPER(ER)-110**: ERα DC_50_ < 3 nM, IC_50_ = 120 nM, **SNIPER(ER)-126**: ERα DC_50_ < 3 nM, IC_50_ = 83 nM). **SNIPER(ER)-110** showed the best ERα degradation efficiency and excellent antitumor activity in the in vivo tumor xenograft model [59].

Astex Pharmaceuticals successfully discovered a nonpeptidomimetic cIAP1 and XIAP inhibitor, **AT-IAP (34)**, through a fragment-based drug discovery using structure information from X-ray crystallography, computational studies, and NMR solution structure analysis. **AT-IAP (34)** showed a strong dual antagonistic efficacy toward XIAP and cIAP1 (XIAP EC_50_ = 5.1 nM, cIAP1 EC_50_ = 0.32 nM). An oral administration of **AT-IAP (34)** in a mouse xenograft model effectively inhibited the tumor growth without affecting the body weight of the mouse [60].

In 2020, GlaxoSmithKline reported a palbociclib-based PROTAC with a CRBN ligand, a VHL ligand, and an IAP-binder (**37**) for the degradation of CDK4 and CDK6. They conjugated three different E3 ligase ligands with palbociclib, an FDA-approved anti-breast-cancer agent, using various linkers. With previously reported CRBN-based CDK4/6-targeting PROTACs [61,62], VHL- and IAP-based PROTACs showed an ability to effectively degrade CDK4/6, (DC_50_ < 10 nM) which could not be achieved with the previously reported VHL- and IAP-based PROTACs [63]. This work again emphasized the importance of linker structures in the development of PROTACs [64]. Moreover, all the Palbociclib-based PROTACs with three different E3 ligases showed the preferential degradation of CDK6 over CDK4 with a marginal degradation efficiency difference.

**37** and **AT-IAP (34)** were also utilized in the development of a series of receptor-interacting serine/threonine protein kinase 2 (RIPK2) PROTACs including **compound 20** and **compound 22** by GlaxoSmithKline (**compound 20**: RIPK2 pDC_50_ = 9.1, **compound 22**: RIPK2 pDC_50_ = 9.8). RIPK2 plays a crucial role in the innate immune system. Therefore, the dysregulation of RIPK2 signaling pathways is highly related to various inflammatory diseases such as inflammatory bowel disease [65], severe pulmonary sarcoidosis [66], and multiple sclerosis [67]. Among the synthesized RIPK2 PROTACs, **compound 20** showed the best profile with excellent solubility, a strong RIPK2 degradation ability, and TNFα inhibition [68].

## 5. MDM2 Ligands

The mouse double minute 2 homolog (MDM2) protein is an E3 ubiquitin ligase that regulates the ubiquitination of p53 and the subsequent proteasomal degradation of p53. In 2004, Roche reported a potent and selective small molecule inhibitor of the MDM2-p53 interaction. They screened a diverse library of synthetic compounds and identified **Nutlin-3 (38)** as a hit compound (Figure 10) [69]. Notably, two enantiomers of *cis*-imidazoline **Nutlin-3 (38)** possessed a highly different binding affinity to MDM2 (IC_50_ of enantiomer a = 13.6 μM, enantiomer b = 0.09 μM). In 2008, the Crews group first reported MDM2-based **PROTAC 14** by conjugating **Nutlin-3 (38)** and a nonsteroidal AR ligand with a PEG linker (Figure 11) [70].

The Sheng group reported a **Nutlin-3** (**38**)-based **homo-PROTAC 11a** that induces the self-degradation of MDM2 to inhibit the MDM2-p53 interaction (DC_50_ = 1.01 μM). **Homo-PROTAC 11a** induced effective MDM2 dimerization and triggered the proteasomal degradation of MDM2 in A549 non-small cell lung cancer cells. In addition, **homo-PROTAC 11a-1**, one of the stereoisomer of **homo-PROTAC 11a**, showed the highest antitumor activity in the A549 xenograft model (IC_50_ = 1.0 μM) [71].

In 2013, Roche optimized the original **Nutlin-3 (38)** compound based on the crystal structure of the p53-MDM2 complex and synthesized a new MDM2 inhibitor, **RG7112 (39)**, with dimethyl substitution on the imidazoline ring and replacement of the methoxy group with a *tert*-butyl group (IC_50_ = 18 nM) [72]. **RG7112** was the first orally available p53-MDM2 inhibitor under clinical trials. In a follow-up study, they discovered a new ligand with an improved affinity based on the crystal structure of MDM2. They replaced the imidazoline structure of **RG7112** with a pyrrolidine moiety and introduced stereochemical configurations for the higher affinity. They reported a second-generation clinical MDM2 inhibitor, **RG7388 (40)**, with an excellent efficacy and selectivity through a SAR study [73] (IC_50_ = 6 nM). In 2020, the Calabretta group reported that **RG7112 (39)**-based PROTAC **YX-2-233** showed a strong degradation of CDK4 and CDK6 in Ph^+^ ALL cells and suppressed S-phase [74]. The Crews group reported that an **RG7388**-based PROTAC, **A1874**, showed a 98% degradation of the BRD4 protein in HCT116 cells with a submicromolar concentration. This was a substantial improvement in target potency compared with their first nutlin-based **PROTAC 14** (**PROTAC 14**: DC_50_ = 10 μM vs. **A1874**: DC_50_ = 32 nM). In addition, A1874 increased p53 stability due to the **RG7388** moiety. The dual mode of action, BRD4 degradation and p53 stabilization, by **A1874** strongly suppressed cancer cell viability compared with VHL ligand-based PROTACs [75].

## 6. DCAF Ligands

Sulfonamide derivatives have drawn attention due to their antibacterial, antifungal, antiviral, and anticancer activities. Recent studies reported that the sulfonamide derivatives **indisulam (41)**, **E7820 (42)**, and chloroquinoxaline sulfonamide **(CQS, 43)** function as a molecular glue that induces the protein–protein interaction between the E3 ligase and the target protein (Figure 12) [76,77].

In 2017, the Nijhawan group revealed that sulfonamide induces the proteasomal degradation of RNA-binding motif protein 39 (RBM39) by interacting with the DCAF15-DDB1-CUL4 complex [76]. In a follow-up study, they investigated how **E7820 (42)** recruits RBM39 to DCAF15 with kinetic analysis and crystal structures (*K*_D_ = 22 μM) [77].

In an independent study, the Owa group reported that sulfonamide induces ubiquitination and the proteasomal degradation of CAPERα through the formation of CAPERα-sulfonamide-DCAF15 [78]. The Chen group reported an **E7820 (42)**-based PROTAC (**DP1**)-targeting BRD4 by employing JQ1 as a target protein ligand (Figure 13). **DP1** showed an excellent degradation of BRD4 in SU-DHL-4 cells (DC_50_ = 10.84 ± 0.92 μM, D_max_ = 98%) and inhibited tumor growth in a mouse model [79].

In 2019, the Cravatt group used a chemical proteomic approach that utilized a FKBP12 ligand, SLF, conjugated with the cysteine-directed electrophilic fragments for the discovery of a nuclear localized E3 ligase. After identifying the electrophilic fragments that induce FKBP12 degradation in the nucleus, they used pull-down-based proteomics and identified DCAF16 E3 ligase as their target protein. One of the electrophilic fragments, **KB02 (44)**, were extensively studied after conjugation with the FKBP ligand (**KB02-SLF**) to monitor its FKBP12 degradation ability (DC_50_ < 2 μM). In addition, they also designed a BRD4-targeting PROTAC, **KB02-JQ1** (DC_50_ ~ 20 μM), by coupling **KB02 (44)** to JQ1, and observed DCAF16 as a valuable nuclear localized E3 ligase [80]. With a similar screening strategy, the Cravatt group also developed electrophilic ligands (**45**) of E3 ligase DCAF11. Using the discovered ligand **45**, they synthesized **21-ARL** for AR degradation with a recruited DCAF11 [81].

## 7. RNF Ligands

In 2019, the Nomura group reported a set of binders to the E3 ligase RNF4 using the ABPP-based covalent ligand screening approach. The optimized covalent ligand **CCW16 (46)** (Figure 14) reacted with one of two zinc-coordinated cysteines in the RING domain without affecting the zinc binding ability of RNF4 (IC_50_ 1.8 μM). They demonstrated a covalent PROTAC **(CCW 28-3)** for BRD4 degradation (Figure 15) [82]. The same group also reported that **Nimbolide (47)**, a natural product exhibiting anticancer activities, was identified as a covalent binder for the E3 ligase RNF114. They used activity-based protein profiling (ABPP) chemoproteomic platforms to discover that **Nimbolide (47)** binds to a cysteine residue of RNF114. Based on the structure of **Nimbolide (47)**, they developed a covalent PROTAC, **XH2**, targeting BRD4 (IC_50_ = 240 nM) [83]. They also demonstrated that **Nimbolide (47)** can be used as a BCR-ABL-targeting PROTAC by recruiting RNF114. They synthesized the degrader **BT1** by coupling the BCR-ABL inhibitor, dasatinib, with **Nimbolide (47)**. They demonstrated that **BT1** selectively degraded BCR-ABL rather than c-ABL, which was also observed in previously reported CRBN or VHL-based BCR-ABL PROTACs [84].

In 2021, they used the same ABPP-based approach to discover the fully synthetic covalent ligand **EN219 (48)**, which targets RNF114 (IC_50_ = 470 nM). The mode of action study showed that **EN219 (48)** mimics the function of a complex natural product, **Nimbolide (47)**. They developed a covalent PROTAC **(ML 2-14)** by conjugating **EN219 (48)** to the BET inhibitor JQ1 for BRD4 degradation [85] (BRD4 _[short]_ DC_50_ = 14 nM, BRD4 _[long]_ DC_50_ = 36 nM).

## 8. AhR Ligands

In 2019, the Naito group developed a novel PROTAC recruiting the aryl hydrocarbon receptor (AhR) E3 ligase complex. They conjugated an AhR ligand **(β-NF, 49)** (Figure 16) with ATRA resulting in a chimeric molecule **β-NF-ATRA** (Figure 17). **β-NF-ATRA**, a PROTAC recruiting CRABPs, induced the degradation of CRABPI and CRABPII in an AhR-dependent manner via the ubiquitin–proteasome pathway [86].

## 9. FEM1B Ligands

The CUL2 E3 ligase FEM1B was recently discovered as an important regulator of the cellular response to reductive stress. In 2022, the Nomura group discovered a chloroacetamide-based covalent ligand, **EN106 (50)** (Figure 16), as a FEM1B ligand (IC_50_ = 2.2 μM). They found the formation of a direct covalent bond between **EN106 (50)** and a cysteine residue on FEM1B by ABPP. They demonstrated that an **EN106**
**(50)**-based PROTAC can be used for the FEM1B recruitment toward target protein degradations. The conjugation of **EN106 (50)** to JQ1 and dasatinib generated **NJH-1-106** (DC_50_ = 250 nM) and **NJH-2-142** and showed the successful degradation of BRD4 and BCR-ABL (Figure 17) [87].

## 10. KEAP1 Ligands

Kelch-like ECH-associated protein-1 (KEAP1) has been known to interact with nuclear factor erythroid 2-related factor-2 (Nrf2) to regulate cellular protective proteins. Therefore, the discovery of the protein–protein interaction inhibitor of KEAP1-Nrf2 has attracted attentions for the treatment of stress-related diseases [88]. In 2020, the Nomura group reported a reversible covalent binding PROTAC using a known KEAP1 ligand bardoxolone methyl **(CDDO-Me, 51)** (Figure 16). They synthesized a bardoxolone-based PROTAC, **CDDO-JQ1** (Figure 17) by conjugating bardoxolone to the BET inhibitor JQ1 (DC_50_ < 100–200 nM). They found that the deletion of a Michael acceptor in **CDDO-JQ1** reduced the BRD4 degradation activity. This indicates that the formation of a reversible covalent bond between the cysteine of KEAP1 and **CDDO-Me (51)** is essential for the recruitment of BRD4 to the KEAP1 E3 ligase [89].

In 2021, the Jin group reported that the E3 ligase KEAP1 can be utilized to develop PROTACs using a highly selective and noncovalent KEAP1 ligand. They developed a KEAP1-recruiting PROTAC, **MS83**, using the previously reported KEAP1 small molecule ligand (**52**, *K*_D_ = 1.3 nM) [90]. **MS83** showed a more durable degradation of BRD3 and BRD4 than **dBET1** in MDA-MB-468 cells (DC_50_ < 500 nM) [91].

In 2022, the Lv group discovered a nature product, Piperlongumine (**PL, 53**), as an E3 ligase ligand. They first confirmed that **PL (53)** was bound to multiple E3 ligases with a competitive ABPP assay. They synthesized PROTAC, **955**, by coupling **PL (53)** with a CDK9 selective inhibitor (SNS-032) and observed the proteasomal degradation of CDK9 by **955** (DC_50_ = 9 nM). Interestingly, they identified that KEAP1 was the only E3 ligase protein recruited by **955** via a covalent attachment of **PL** using the TurboID-bait assay [92]. This observation suggested that E3 ligase selectivity could be changed after the conjugation of the E3 ligase ligand to a target protein ligand.

## 11. Conclusions

PROTAC has been developed as a new strategy for disease treatment in the chemical biology community over the past 20 years. PROTACs harness an intracellular proteolytic system for the degradation of a POI. This strategy has emerged as an alternative to overcome the limitations of conventional drug discovery by targeting undruggable proteins. For effective TPD, the selection of E3 ligase ligands and target protein ligands is critical for PROTAC design. Although a number of ligands for various proteins have been reported during the traditional drug discovery campaigns so far, only a few E3 ligase ligands are currently available for TPD. For those reasons, most of the PROTAC research has focused on the demonstration of the TPD concept for the various drug target proteins using CRBN or VHL ligands during the last decades. However, the desired target protein degradations often are compromised due to the cell type or tissue type dependent expression profiles of CRBN or VHL. Moreover, resistance to CRBN- or VHL-based PROTACs has been recently observed. Therefore, the discovery of ligands for new E3 ligases has drawn attention as they can be used for effective PROTAC development. In addition, there is still a lack of understanding of the previously developed PROTACs. For example, a given drug molecule connected to different E3 ligase ligands has shown to exhibit different PROTAC efficiencies [45,74], target selectivities [64], and drug resistance profiles in various cancer cells [4]. Although there is a report on the systematic approach taken to select the E3 ligase ligand in PROTAC designs, it is restricted to the kinase degraders and only three E3 ligase ligands were studied [93]. Their kinase-targeting PROTACs mostly prefer CRBN ligands over VHL or IAP ligands for the efficient protein degradation. Additional studies on other protein families are needed for a comprehensive understanding of E3 ligase ligand selection for efficient protein degradation. This suggests that the ternary complexes that are recruited by bifunctional PROTACs could be susceptible to ligand orientation, the ligand’s hydrophobicity, and the physicochemical properties of the chemical linkers. Further studies related to various ternary complexes should provide clearer pictures on future PROTAC designs.

Herein, we reviewed the discovery of various E3 ligase ligands and their applications for the design of PROTACs. Considering more than 600 types of E3 ligases, the development of ligands for various E3 ligases would involve the expansion of the toolbox in TPD to overcome the current limitation of PROTACs. Additionally, the cell or tissue type specific unexplored E3 ligases would be the basis for a new type of PROTAC, which could control a certain protein in a spatially specific way. The spatial protein degradation in lesions would treat diseases very effectively without toxicity or side effects in normal tissues. Therefore, the discovery of novel E3 ligase ligands would be an important goal to expand the landscape of PROTACs toward promising therapeutics in the future.

## Figures and Tables

**Figure 1 molecules-27-06515-f001:**
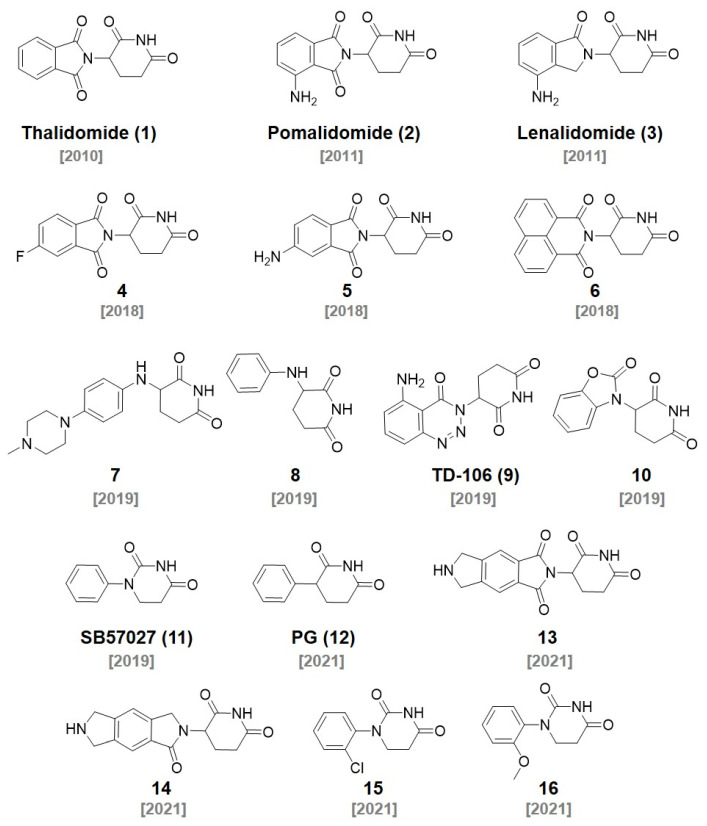
Chemical structures of CRBN ligands and their derivatives.

**Figure 2 molecules-27-06515-f002:**
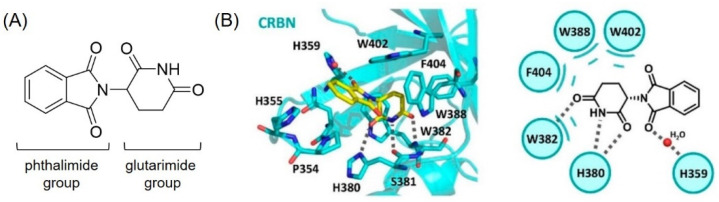
(**A**) Chemical structure of thalidomide composed of phthalimide and glutarimide groups. (**B**) Schematic illustrations of thalidomide binding mode to CRBN (PDB: 4CI1). Reprinted with permission from ref. [14] (Copyright 2021 American Chemical Society).

**Figure 3 molecules-27-06515-f003:**
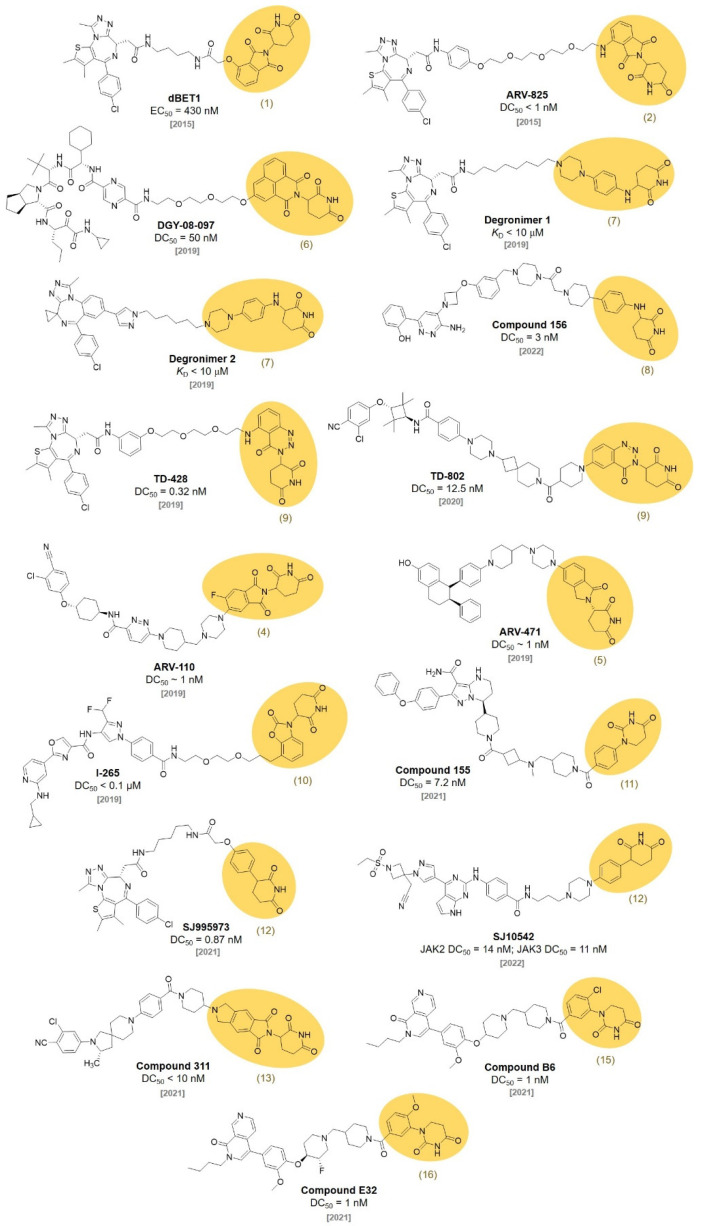
Examples of CRBN ligand-based PROTACs.

**Figure 7 molecules-27-06515-f007:**
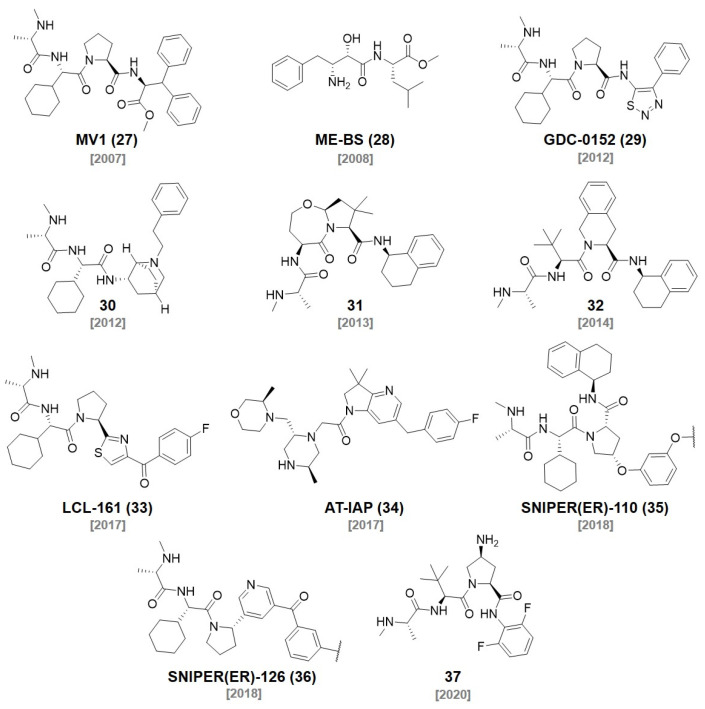
Representative examples of IAP ligands.

**Figure 8 molecules-27-06515-f008:**
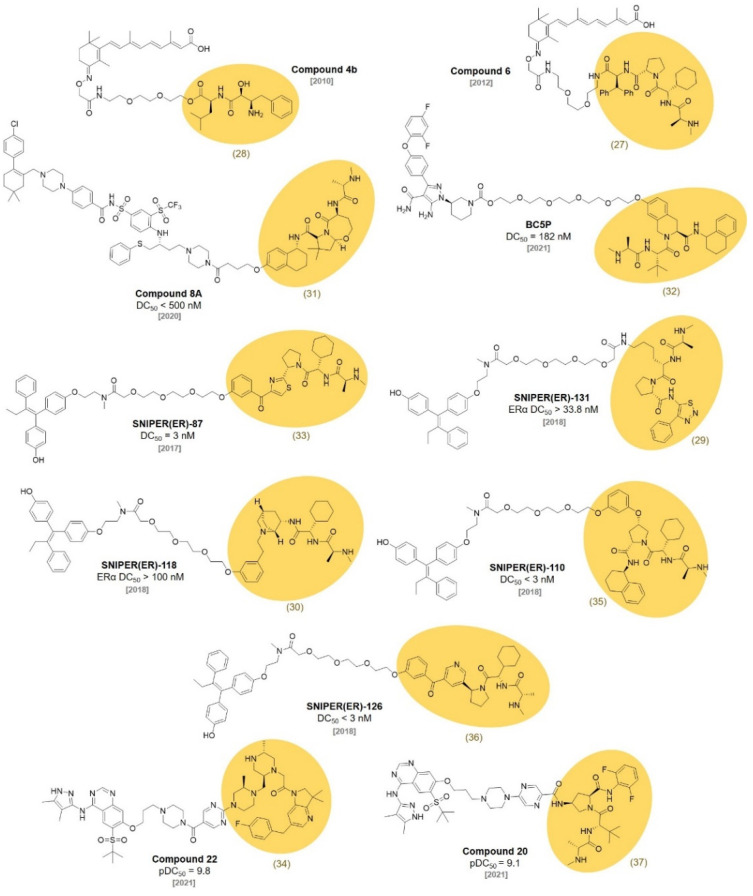
Selected examples of IAP ligand-based PROTACs.

**Figure 9 molecules-27-06515-f009:**
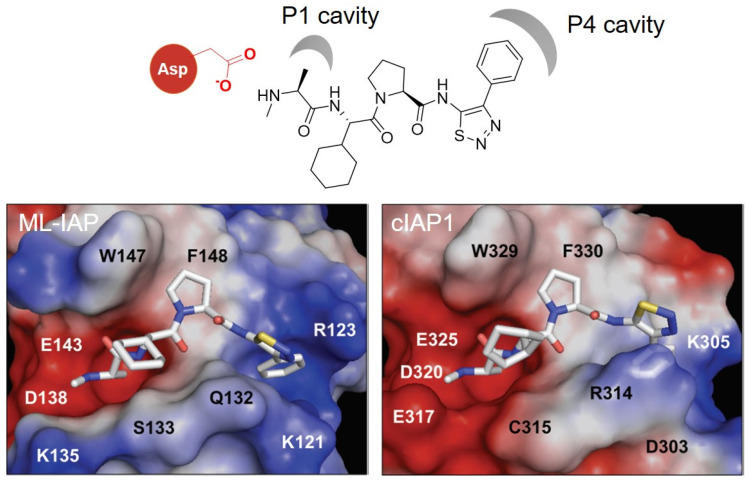
Chemical structure of **GDC-0152** and its key binding modes to ML-IAP (**left**) and cIAP1 (**right**). Reprinted with permission from ref. [53] (Copyright 2012 American Chemical Society).

**Figure 10 molecules-27-06515-f010:**
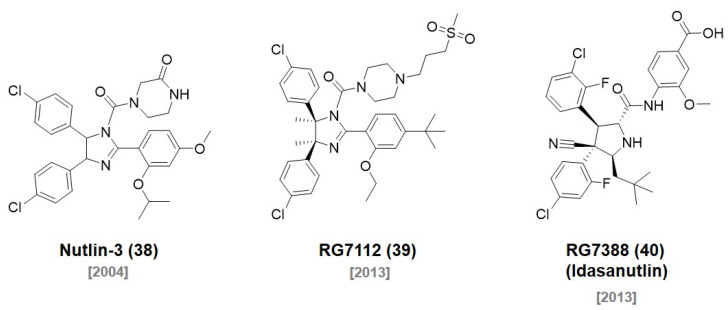
Chemical structures of reported MDM2 ligands.

**Figure 11 molecules-27-06515-f011:**
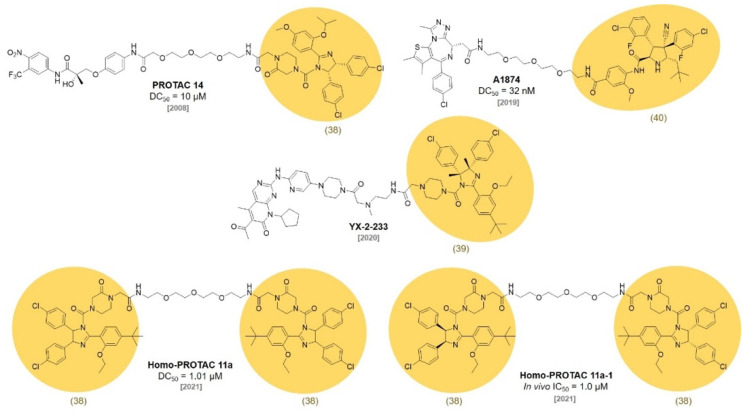
Representative examples MDM2 ligand-based PROTACs.

**Figure 12 molecules-27-06515-f012:**
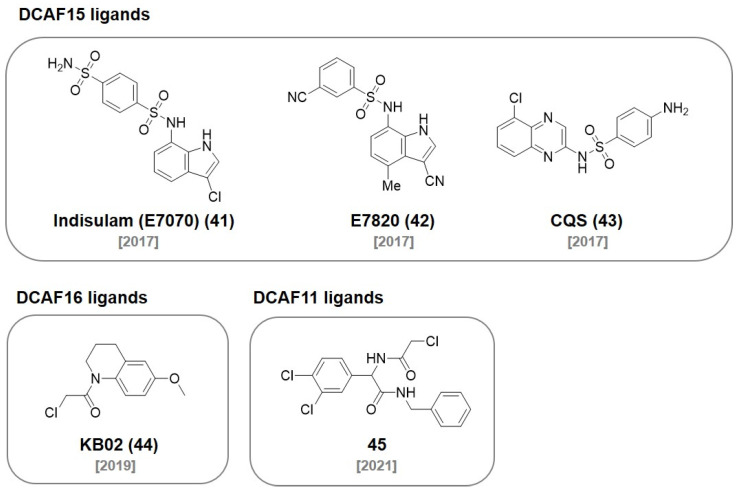
Chemical structures of recently reported DCAF ligands.

**Figure 13 molecules-27-06515-f013:**
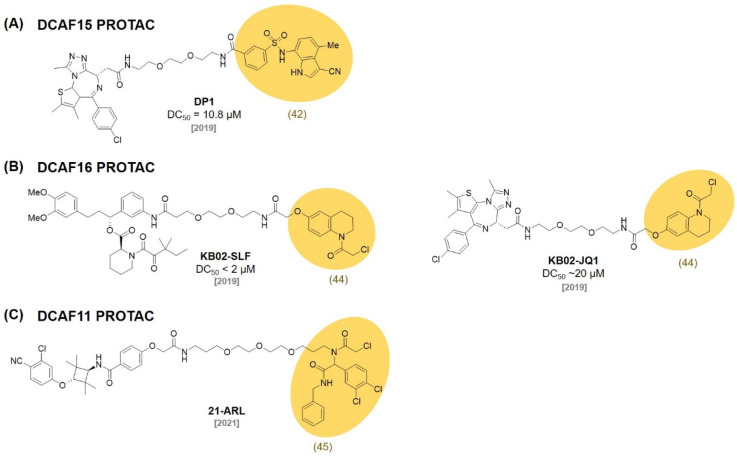
Recently developed PROTACs utilizing various DCAF ligands.

**Figure 14 molecules-27-06515-f014:**
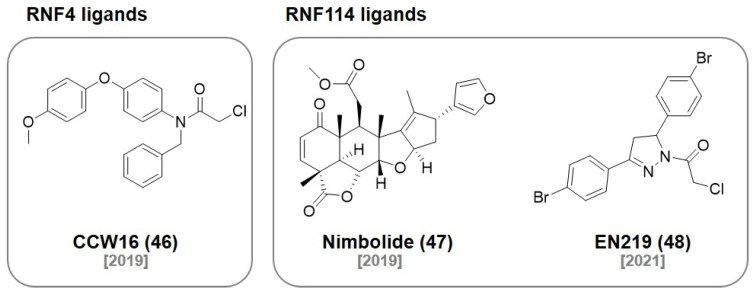
Recent examples of RNF ligands.

**Figure 15 molecules-27-06515-f015:**
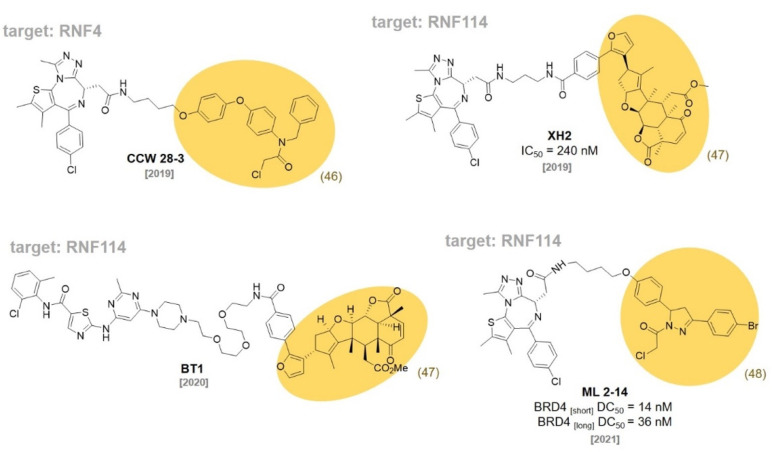
RNF ligand-based PROTACs targeting RNF4 and RNF114.

**Figure 16 molecules-27-06515-f016:**
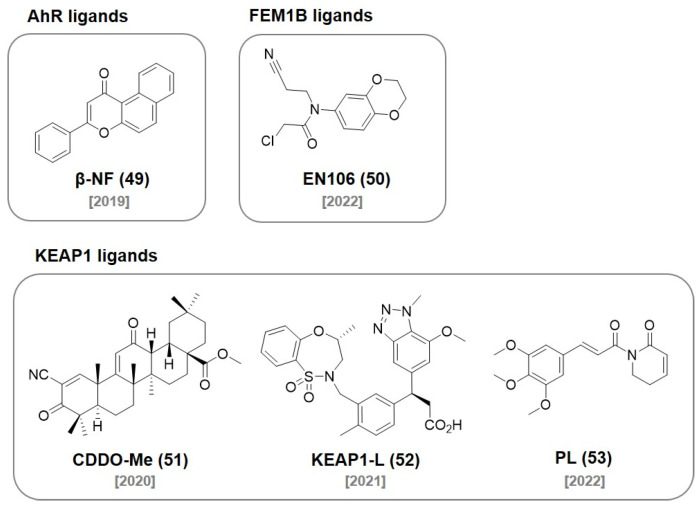
Recently developed **AhR**, **FEM1B**, and **KEAP1** ligands.

**Figure 17 molecules-27-06515-f017:**
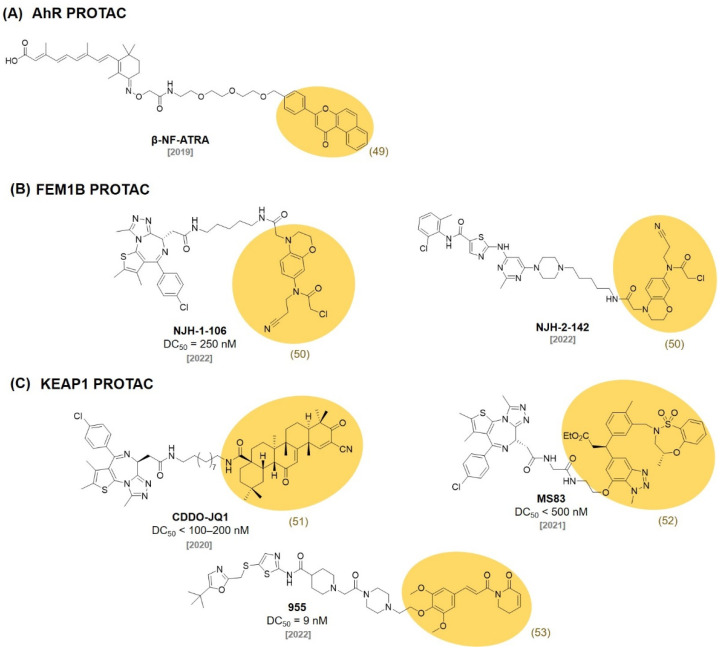
Chemical structures of PROTACs using (**A**) an AhR ligand, (**B**) FEM1B ligands, and (**C**) KEAP1 ligands.

## Data Availability

Not applicable.

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
