# Peer review of "Discovery of E3 Ligase Ligands for Target Protein Degradation"

_molecules, 2022, doi:10.3390/molecules27196515_

Round 1
Reviewer 1 Report
This paper reviewed the ligands of different E3 ligases used for PROTACs development. The representative PROTACs with these E3 ligase ligands were also discussed. The topic is very interesting and useful for the discovery and development of PROTACs. In generally, the manuscript is well-organized and the references can support the conclusions. The key issues should be addressed before its publication on Molecules.
1. The authors are suggested to discuss the crystal structures of these the E3 ligases in complex with their ligands (if available) to give more ideas about the possible tethering positions of these linkers at the ligands for the successful PROTACs design.
2. The Kd values (or the other binding affinity parameters; if available) of these E3 ligase ligands should be included in the manuscript, which will be very useful for the design and development of new PROTACs.
3. The protein degradation activities (such as DC50, if available) of these representative PROTACs are suggested to be included together with the structures in the Figures to give more ideas to the readers about the different potentials of these E3 ligase ligands.
4. The expression levels of these different E3 ligases involving the current PROTACs development (if reported) are suggested to be discussed.
5. For many protein targets, the related PROTACs with ligands of different E3 ligases were reported. The authors are suggested to compare and discuss their activities.
Reviewer 2 Report
This is an extremely interesting paper reviewing all E3 ligase ligands has been discovered to improve the current PROTAC technology and its application. This kind of review was still missing in the international literature, therefore, I believe that it will be really appreciated from worldwide researchers.
Minor Points:
However, I have some suggestion to improve the quality of the paper.
Since the review the review retraces a very well detailed historical excursus, I believe that authors should mention the following references in the introduction (I suggest after 47-48 lines):
- Roth, S.; Fulcher, L.J.; Sapkota, G.P. Advances in targeted degradation of endogenous proteins. Cell. Mol. Life Sci. 2019, 76, 2761–2777.
- Prozzillo, Y.; Fattorini, G.; Santopietro, M.V.; Suglia, L.; Ruggiero, A.; Ferreri, D.; Messina, G. Targeted Protein Degradation Tools: Overview and Future Perspectives. Biology 2020, 9, 421. https://doi.org/10.3390/biology9120421
- Nabet, B.; Roberts, J.M.; Buckley, D.L.; Paulk, J.; Dastjerdi, S.; Yang, A.; Leggett, A.L.; Erb, M.A.; Lawlor, M.A.; Souza, A.; et al. The dTAG system for immediate and target-specific protein degradation. Nat. Chem. Biol. 2018, 14, 431–441.
Reviewer 3 Report
This is an outstanding review on E3-based Protac developments. The work describes with high detail the ligands developed to interact with E3-ligases and how they have been conjugated to different linkers and warheads to create Protacs in the last two decades. It is clearly written and references are correct.
Due to the complexity of Protacs, the manuscript will be an important compass to readers to navigate in this topic.
I think that the paper should accepted with minor corrections.
Minor points:
-line 35 : “ Considering every protein in cell is essential for the cellular homeostasis “ : this sounds like an overstatement. This sentence should be reformulated.
-line 52 : “Those PROTACs showed the limitations due to the cellular-specific expression of E3 ligases and the emergence of drug resistance toward PROTACs having CRBN or VHL ligands". Could authors give specific examples of that?
-line 62: when defining cereblon ligands, I couldn’t find any paper from B. Ebert laboratory. Given the fact that Ebert’s lab has made pivotal contributions, I suggest to include at least one reference from the group. If not, authors should justify the reason why.
- Line 79 : “enatioselective” should be “enantioselective”
-line 131 : “theses” shoud be “these”
-lines 159-168 : I couldn’t find the figures of compounds 311, B6 and E32.
- line 159 : could the paragraph be referenced?
-Figures: there are inconsistencies un the numbering of compounds in parenthesis: (47), (48),….. specially because Figure 11 and 13 have been introduced later, probably.
